# Resistin is a risk factor for all-cause mortality in elderly Finnish population: A prospective study in the OPERA cohort

Karri Parkkila[1], Antti Kiviniemi [1], Mikko Tulppo[2], Juha Perkiömäki[1], Y. Antero Kesäniemi[1], Olavi Ukkola [1]*

1 Medical Research Center Oulu, Oulu University Hospital, Research Unit of Internal Medicine, University of Oulu, Oulu, Finland, 2 Department of Physiology, Research Unit of Biomedicine, Faculty of Medicine, University of Oulu, Oulu, Finland

* olavi.ukkola@oulu.fi

**Data Availability Statement:** Data cannot be shared publicly because of privacy policy. Data are available from the Oulu University Institutional Data Access for researchers who meet the criteria for

## Abstract

### Objective

Resistin is a small, cysteine-rich proinflammatory molecule that is primarily secreted by peripheral blood mononuclear cells and macrophages in humans. Previous studies have shown resistin to participate in various pathological processes including atherosclerosis and cancer progression but not many studies have assessed the role of resistin as a risk factor for all-cause mortality. The objective of this prospective study was to evaluate whether resistin predicts mortality among elderly Finnish people.

### Methods

The study population consisted of 599 elderly (71.7 ± 5.4 years) patients and the follow-up was approximately six years. A thorough clinical examination including anthropometric and other clinical measurements such as blood pressure as well as various laboratory parameters (including resistin) was conducted at baseline.

### Results

After the follow-up, 65 (11%) of the patients died. Resistin was a significant risk factor for all-cause mortality (HR 3.02, 95% CI: 1.64–5.56, p<0.001) when the highest tertile was compared to the lowest. Resistin remained as a significant risk factor even after adjusting for various covariates such as age, sex, systolic blood pressure, smoking habits, alcohol consumption, medications (antihypertensive, lipid-lowering, glucose-lowering), hsCRP and leisure time physical activity. Receiver operating characteristic (ROC) curve analysis for resistin demonstrated area under the curve (AUC) of 0.656 (95% CI: 0.577–0.734), p<0.001 and an optimal cutoff value of 12.88 ng/ml.

### Conclusions

Our results indicate that resistin is a significant risk factor for all-cause mortality among elderly Finnish subjects, independent from traditional cardiovascular risk factors.

access to confidential data. Contact information for the Oulu University Institutional Data Access committee: olavi.ukkola@oulu.fi, tel. +358400944539

**Funding:** The authors received no specific funding for this work.

**Competing interests:** The authors have declared that no competing interests exist.

## Introduction

Resistin, a small secreted 12.5 kDa cysteine-rich molecule, is the founding member of resistin-like molecules with hormone-like activity [1]. In humans, the primary source of resistin are peripheral blood mononuclear cells, macrophages and bone marrow cells [2]. Resistin is a pro-inflammatory molecule [3] and high plasma resistin levels are associated with inflammatory markers such as highly sensitive C-reactive protein (hsCRP) and leucocytes [4]. Moreover, Lehrke et al. [5] reported that resistin secretion is incited by pro-inflammatory cytokines like interleukin (IL) 6 and tumor necrosis factor alpha (TNFα). Therefore, resistin could mediate the inflammatory effects on arterial wall and contribute to the development of atherosclerosis. The latter notion is supported by the earlier data showing that high resistin level is a risk factor for cardiovascular events but when adjusting for inflammatory markers, this association is no longer significant [6]. In contrast, Butler et al. [7] reported that resistin was a significant risk factor for new onset heart failure even when adjusting for various inflammation markers (CRP, IL-6, TNFα). Therefore, some other mechanisms not related to the inflammation should also be considered.

Since its discovery in 2001 [8], the biological effects of resistin and its association with various pathological processes such as insulin resistance [9], vascular smooth muscle cell proliferation [10], endothelial dysfunction, reduced nitric oxide (NO) bioavailability [11] and atherosclerosis [12] has been extensively investigated. However, fewer studies have addressed the association between resistin concentration and all-cause mortality. A recent systematic review and meta-analysis including eight prospective studies with elderly patients and with an average follow-up of 1.0 to 7.8 years showed that resistin was a significant risk factor for all-cause and cardiovascular mortality [13]. It should be noted, however, that the study populations consisted of high-risk patients with conditions like coronary artery disease, myocardial infarction, type 2 diabetes mellitus, ischemic stroke or end stage renal disease.

Resistin has also been proposed as a molecular link between aging and age-related conditions [14]. However, there is hardly any previous research exploring the role of resistin as a risk factor for all-cause mortality among elderly patients representing the general population. A multi-ethnic study conducted in the United States with study population of Caucasian, Chinese, Black and Hispanic ethnicities (age varying from 45 to 84) and without clinically apparent cardiovascular disease showed that resistin was a significant risk factor for cardiovascular events but not for all-cause mortality [15]. The aim of our study was to elucidate the interaction between resistin and all-cause mortality among relatively healthy elderly population and therefore broaden the current understanding regarding resistin as a biomarker of aging and related mortality.

## Methods

### Study population

This research was conducted as part of the OPERA (Oulu Project Elucidating Risk for Atherosclerosis) project. The initial study population recruited during the first phase (between 1991 and 1993) consisted of middle-aged patients with hypertension (n = 519) and their age- and sex-matched controls (n = 526). Of the 813 survivors, 600 (62–83 years of age) attended a follow-up visit between 2013 and 2014. The original study cohorts and selection criteria have been described in detail previously [16]. The OPERA study was approved by the Ethical Committee of the Faculty of Medicine, University of Oulu and was compatible with the Declaration of Helsinki. Informed consent was obtained from each participant.

The study population in the current study composes of those 600 patients who attended the follow-up visit during 2013–2014. A thorough clinical examination including weight, height,

waist and hip as well as blood pressure and heart rate measurements were conducted at that visit. Blood pressure was measured according to the recommendations of the American Society of Hypertension [17] in a sitting position from the right arm with an oscillometric device (Dinamap® model 18465X, Criticon Ltd., Ascot, UK) after an overnight fast and after a 10-15-minute rest. Three measurements were made at 1-minute intervals and the means of the last two were used in the analyses. Hypertension was defined as blood pressure over 140/90 or current antihypertensive medication. Diabetes was defined as a known diabetes mellitus diagnosis (fasting plasma glucose of 7.0 mmol/l or more or 2-h plasma glucose of 11.1 mmol/l or more in the oral glucose tolerance test) at baseline visit. Body mass index (BMI) was calculated as weight (kg) divided by squared height ($m^2$). The Mini-Mental State Examination (MMSE) is an easy-to-use tool for evaluating one's memory and the processing of information. MMSE consists of simple questions and tasks, and correct answers accumulate points (the maximum being 30). The lower the score, the worse the memory function. Likewise, the EQ-5D is a standardized questionnaire for assessing one's overall health status. It consists of two parts: descriptive and evaluation. In the evaluation part, the patients rate their overall health status using the visual analogue scale (ranging from 0 to 100), respectively. The higher the score, the better the experienced health status. The scores of the EQ-5D evaluation part are presented in the current study. Additionally, a questionnaire regarding detailed information about smoking habits, alcohol consumption, physical activity, medication and previous medical history was introduced to each participant.

## Laboratory analyses

The laboratory tests were carried out in Joint Municipal Service Provider of Northern Finland Laboratory Center, NordLab Oulu (after 12 h fasting) using Siemens Advia 1800 chemistry and Siemens Advia Centaur XP immunochemistry analyzers (Siemens Healthcare Diagnostics Oy). Total-, high-density lipoprotein (HDL)- and low-density lipoprotein (LDL) cholesterol and triglycerides were analyzed by enzymatic methods. Plasma resistin levels were measured according to manufacturer's instructions by using a commercially available enzyme-linked immunosorbent assay kit (Millipore Corporation, Billerica, Massachusetts) with intra- and interassay coefficients of variation of 3.2–7.0 and 7.1–7.7 respectively. Plasma high-sensitivity C-reactive protein (hsCRP) concentration was measured using the commercially available ELISA kit (Diagnostic Systems Laboratories).

## Outcome classification

After the baseline clinical examinations and laboratory tests between 2013 and 2014, the participants were followed-up on average 67 ± 12 months (the end of follow-up being death or the last day of 2019). Information on causes of death and events leading to hospitalization was obtained from Finnish Causes-of-Death Register and the Hospital Discharge Register.

## Statistical methods

Out of 600 patients, the resistin values were missing from one patient, so the final study population consisted of 599 participants. Patients were divided into tertiles based on their resistin concentrations. The differences in baseline characteristics across groups were compared using one-way ANOVA + Bonferroni post hoc test (normally distributed variables), Kruskal-Wallis + Mann-Whitney U-test (non-normally distributed variables) and Chi-square test + post hoc analyses (categorical variables). Time-to-death data was analyzed using Cox proportional hazards models. Due to heavily skewed distributions, resistin and hsCRP were analyzed as log-transformed. Resistin was first analyzed as continuous (log-transformed) and then additional

analyses were performed by analyzing resistin as categorical (tertiles) variable. The effect of confounding factors was controlled by adding them into the multivariate model. After univariate analysis, the following statistically significant risk factors (in addition to resistin) for all-cause mortality were added into in the multivariate Cox regression analysis: age, antihypertensive medication, log-hsCRP and physical activity. Even though not statistically significant risk factors for all-cause mortality, gender, systolic blood pressure, medications (lipid-lowering and glucose-lowering), smoking habits and alcohol consumption were added into the multivariate model in addition to aforementioned covariates, based on their generally acknowledged role as cardiovascular risk factors. The discrimination abilities of resistin were assessed by conducting receiver operating characteristics (ROC) curve analysis. The optimal cutoff for resistin was determined based on the ROC curve and the cutoff value was further used in survival analyses. Statistical analyses were performed using IBM SPSS version 26. Statistical significance was set at $p < 0.05$.

## Results

The baseline characteristics of the study population are presented in Table 1. Our study population consisted of 599 patients in total, of which 281 (47%) were men. At baseline, the average age of our patients was $71.7 \pm 5.4$. When comparing the participants in the highest resistin tertile to participants in the lowest, those who were in highest tertile were older ($p < 0.001$), suffered more deaths ($p < 0.001$), had higher BMI ($p = 0.025$) and hsCRP concentration ($p < 0.001$). Those in the highest tertile had lower diastolic blood pressure ($p = 0.002$) as well as lower HDL cholesterol ($p = 0.024$), and a larger proportion of them reported never consuming alcohol compared to those in the lowest tertile ($p = 0.021$). 75% of our patients had hypertension and 28% presented diabetes at baseline. Over three quarters of the patients (78%) were taking antihypertensive or cardiac medications and nearly half of the patients (48%) were using lipid medications. In a questionnaire about leisure time physical activity, only 7.2% of the patients reported not engaging in any physical activity whereas 69% described participating in regular or frequent exercising. The participants in the highest resistin tertile reported more often to not engage in any physical activity compared to those in the lowest tertile ($p = 0.004$).

### All-cause mortality

During an average follow-up period of $67 \pm 12$ months, 65 (11%) of the patients died. Risk factors for all-cause mortality were, as expected, age (HR 1.15, 95% CI: 1.09–1.20, $p < 0.001$), hypertension (HR 2.19, 95% CI: 1.08–4.43, $p = 0.029$, compared to non-hypertensive), diabetes (HR 1.79, 95% CI: 1.09–2.94, $p = 0.022$, compared to non-diabetic), log-hsCRP (HR 1.04, 95% CI: 1.01–1.06, $p = 0.006$) and self-reported leisure time inactivity (HR 3.61, 95% CI: 1.86–7.03, compared to those who reported frequently exercising). Surprisingly, traditional cardiovascular risk factors such as gender, systolic blood pressure, alcohol consumption or smoking habits were not statistically significant risk factors for all-cause mortality during the follow-up period.

### Resistin and all-cause mortality

At baseline, the median resistin concentration ($1^{st}$-$3^{rd}$ quartile) was 10.2 (8.09–12.7) ng/ml and varied from 2.94 to 43.1 ng/ml. Due to highly skewed distribution, resistin was analyzed as a log-transformed variable in Cox regression analyses. Resistin was a significant risk factor for all-cause mortality when analyzed as a continuous variable (HR: 1.089, 95% CI: 1.056–1.123, $p < 0.001$), log-transformed (HR: 23.051, 95% CI: 5.778–91.958, $p < 0.001$), as well as when the highest and lowest tertiles were compared (HR 3.020, 95% CI: 1.639–5.564, $p < 0.001$). The

**Table 1. Characteristics of the study population.** Patients are divided into tertiles based on their resistin concentrations.

| Variable | 1st tertile | 2nd tertile | 3rd tertile | p-value |
|---|---|---|---|---|
| Number of patients | 200 | 200 | 199 | |
| Deaths | 14 (7%) | 12 (6%) | 39 (19.6%) | <0.001* |
| Males | 93 (47%) | 97 (49%) | 91 (46%) | 0.849 |
| Age (years) | 69.8 ± 5.0 | 71.6 ± 5.1 | 73.7 ± 5.5 | <0.001* |
| BMI (kg/m$^2$) | 27.8 (25.1–31.6) | 28.3 (25.3–31.6) | 28.8 (26.1–32.7) | 0.025* |
| SBP (mmHg) | 139 ± 22 | 139 ± 22 | 137 ± 22 | 0.745 |
| DBP (mmHg) | 74 ± 9.8 | 73 ± 10 | 70 ± 11 | 0.002* |
| Total cholesterol (mmol/l) | 4.79 ± 1.07 | 4.76 ± 1.04 | 4.62 ± 1.01 | 0.191 |
| HDL (mmol/l) | 1.45 (1.20–1.79) | 1.40 (1.19–1.75) | 1.35 (1.09–1.67) | 0.022* |
| LDL (mmol/l) | 2.88 ± 0.96 | 2.87 ± 0.98 | 2.76 ± 0.94 | 0.359 |
| Triglycerides (mmol/l) | 1.13 (0.92–1.57) | 1.11 (0.90–1.48) | 1.25 (0.98–1.75) | 0.009* |
| Resistin (ng/ml) | 7.27 (6.23–8.09) | 10.2 (9.54–10.9) | 14.4 (12.7–16.7) | <0.001* |
| hsCRP (mg/l) | 0.90 (0.50–2.08) | 1.30 (0.70–3.15) | 1.90 (1.00–4.20) | <0.001* |
| Smoking habits | | | | 0.083 |
| Never | 106 (53%) | 92 (46%) | 108 (54%) | |
| Current smoker | 12 (6%) | 16 (8%) | 10 (5%) | |
| Not anymore | 76 (38%) | 90 (45%) | 81 (41%) | |
| Irregularly | 6 (3%) | 2 (1%) | 0 (0%) | |
| Alcohol consumption | | | | 0.024* |
| Never | 28 (14%) | 27 (14%) | 48 (24%)* | |
| Not anymore | 22 (11%) | 19 (9.5%) | 26 (13%) | |
| Occasionally | 117 (59%) | 130 (65%) | 107 (54%) | |
| More than once a week | 27 (14%) | 21 (11%) | 17 (8.5%) | |
| Almost daily | 6 (3%) | 3 (1.5%) | 1 (0.5%) | |
| Hypertension | 142 (71%) | 148 (74%) | 159 (80%) | 0.113 |
| Diabetes | 44 (22%) | 55 (28%) | 67 (34%) | 0.034* |
| Medication | | | | |
| Antihypertensive/cardiac | 143 (72%)* | 155 (78%) | 167 (84%)* | 0.012* |
| Lipid-lowering | 100 (50%) | 93 (47%) | 97 (49%) | 0.777 |
| Glucose-lowering | 42 (21%) | 51 (26%) | 57 (29%) | 0.208 |
| Leisure time physical activity | | | | 0.014* |
| No physical activity | 9 (4.5%) | 10 (5%) | 24 (12%)* | |
| Irregular exercise | 45 (23%) | 52 (26%) | 43 (22%) | |
| Regular exercise | 38 (19%) | 43 (22%) | 50 (25%) | |
| Frequent exercise | 108 (54%) | 95 (48%) | 82 (41%) | |
| EQ-5D (0–100) | 80 (75–90) | 80 (70–90) | 80 (65–90) | 0.010* |
| MMSE (0–30) | 28 (27–29) | 28 (26–29) | 28 (26–29) | 0.297 |

The values are mean (± SD), median (1st-3rd quartile) or number of patients (% of the group). The groups were compared using one-way ANOVA + Bonferroni post hoc (normally distributed variables), Kruskal-Wallis + Mann-Whitney U-test (skewed distributions) and Chi-square + post hoc analyses (categorical variables). *Statistically significant (p<0.05) difference between the groups.

ROC curve analysis (Fig 1) showed the optimal cutoff concentration for resistin to be 12.88 ng/ml, and those who had resistin concentration above the cutoff had over 3-fold risk for all-cause mortality compared to those below the cutoff (p<0.001). The effect of resistin on mortality persisted after adjusting for age, sex, systolic blood pressure, smoking habits, alcohol consumption, medications (antihypertensive/cardiac, lipid-lowering and glucose-lowering), log-

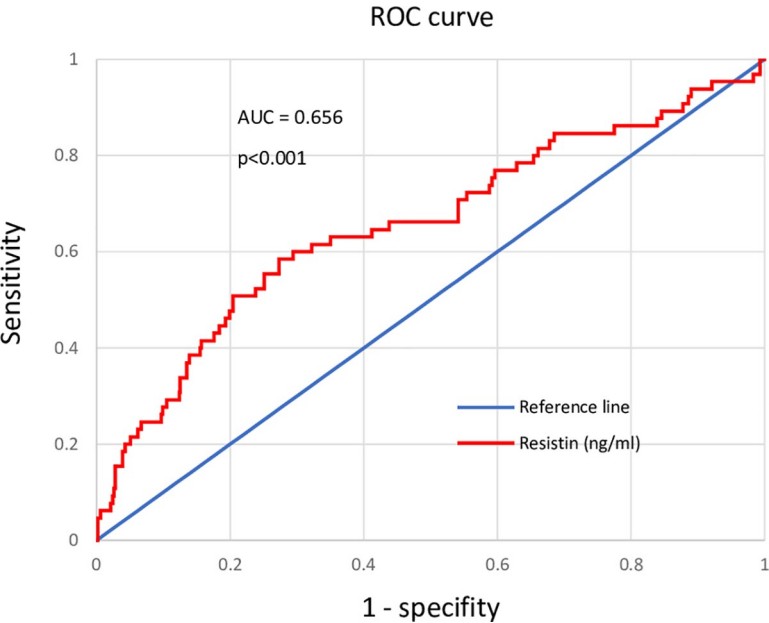

**Fig 1. Receiver operating characteristic (ROC) curve for resistin in predicting all-cause mortality.** ROC was performed for plasma resistin concentration (n = 599). During the follow-up period of 67 ± 12 months, 65 deaths were recorded. Area under the curve (AUC) for resistin 0.656 (95% CI: 0.577–0.734), p<0.001.

hsCRP, and leisure time physical activity (Table 2). After subgroup multivariate analyses, resistin remained as a significant predictive factor for all-cause mortality only in the following subgroups: highest age tertile (p = 0.001), those who had hypertension (p = 0.038), those who had diabetes (p = 0.002) and those who had ever used alcohol (p = 0.006). The cumulative hazard for each resistin tertile is presented in Fig 2.

## Discussion

Our findings suggest that high plasma resistin concentration is a significant risk factor for all-cause mortality among elderly patients. The association between resistin and all-cause mortality persisted after adjusting for age, sex, systolic blood pressure, smoking habits, alcohol consumption, medications (antihypertensive/cardiac, lipid-lowering and glucose-lowering),

**Table 2. Hazard ratios for resistin in predicting all-cause mortality.**

| Resistin | Univariate | | Multivariate | |
|---|---|---|---|---|
| | Hazard ratio (95% CI) | p-value | Hazard ratio (95% CI) | p-value |
| Continuous | 1.089 (1.056–1.123) | <0.001 | 1.058 (1.014–1.105) | 0.010 |
| log-transformed | 23.051 (5.778–91.958) | <0.001 | 5.482 (1.103–27.248) | 0.038 |
| 1st tertile | Reference | <0.001 | Reference | 0.006 |
| 2st tertile | 0.852 (0.394–1.841) | 0.683 | 0.626 (0.281–1.394) | 0.251 |
| 3rd tertile | 3.020 (1.639–5.564) | <0.001 | 1.801 (0.916–3.538) | 0.088 |
| ≥ 12.88 ng/ml (cutoff) | 3.649 (2.243–5.936) | <0.001 | 2.750 (1.640–4.612) | <0.001 |

Multivariate Cox regression analyses adjusted for age, sex, systolic blood pressure, smoking habits, alcohol consumption, medications (glucose-lowering, antihypertensive and lipid-lowering), high-sensitivity CRP as well as leisure time physical activity. Due to skewed distributions, resistin and hsCRP were analyzed as log-transformed.

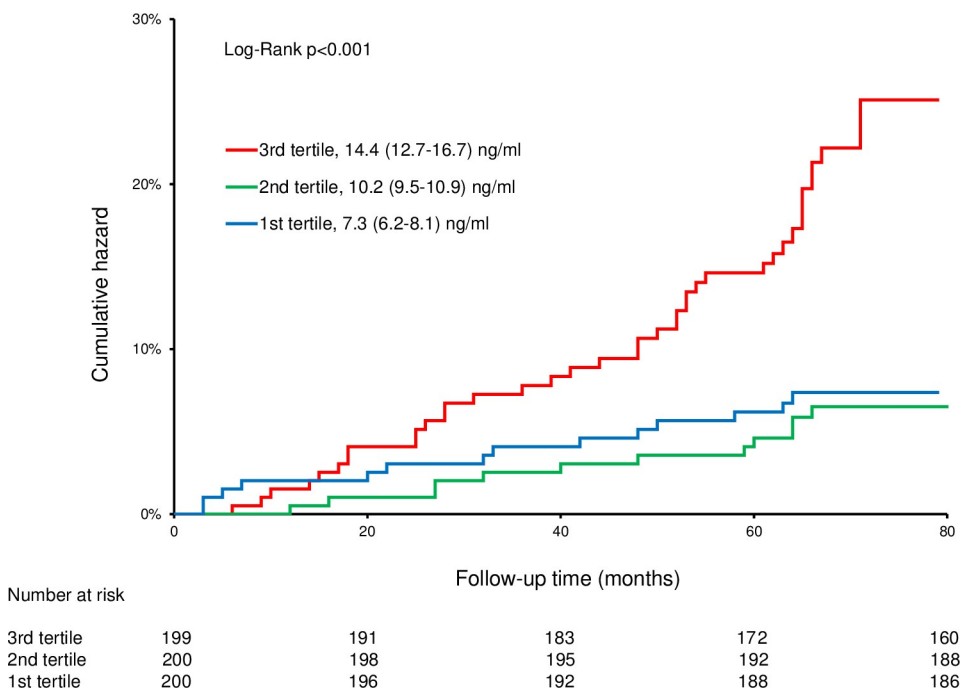

**Fig 2. Increased resistin concentration predicts all-cause mortality.** Kaplan-Meier curves present the cumulative hazards of each resistin tertile. Median (1st-3rd quartile) resistin concentrations for each tertile are presented.

hsCRP, and leisure time physical activity. Therefore, resistin seems to predict all-cause mortality independent from traditional cardiovascular risk factors.

In the current study, we report that resistin is a significant risk factor for all-cause mortality among elderly participants. Agreeing with our findings, a recent systematic review and meta-analysis consisting of seven prospective studies concluded that resistin is a significant risk factor for all-cause mortality [13]. The participants in the meta-analysis, however, were suffering from various chronic conditions such as coronary artery disease, ischemic stroke, myocardial infarction, type 2 diabetes and end stage renal disease whereas our study represents more of a general population. In addition, the baseline resistin concentrations varied significantly between the included studies in the aforementioned meta-analysis (3.6–127.2 ng/ml). Similarly, there is evidence on the association between increasing resistin concentration and mortality in critically ill patients with sepsis [18] and traumatic brain injury [19], for instance.

However, fewer studies have explored the prospective role of resistin in healthier populations. Muse et al. [15] conducted a prospective multi-ethnic (Caucasian, Chinese, Hispanic, African American) study in the United States with a total of 1913 patients without clinically apparent cardiovascular diseases (age varying from 45 to 84) with a follow-up of 7.2 ± 1.8 years. On the contrary to our results, they documented that resistin was not a significant risk factor for all-cause mortality after adjusting for age, sex and race. Interestingly, they reported that there was a significant difference between ethnicities in resistin concentrations with African Americans representing the highest and Chinese the lowest levels at baseline. The differences between Muse et al. and our study are likely due to differences in the age spectrums and diversities of the study populations between the studies. Gencer et al. [6] conducted a prospective study in The Health, Aging and Body Composition cohort which included similar study population as in our study. They reported that resistin was significantly associated with cardiovascular events but adjusting for inflammatory markers markedly attenuated this effect. Our

findings, on the other hand, indicate that resistin predicts all-cause mortality independent from inflammatory markers, albeit the differences in outcome variables between our studies need to be considered.

There is a rich body of evidence showing the role of resistin as a mediator of the inflammatory process. We have also reported earlier using the same cohort as in the present study that high plasma resistin levels are associated with enhanced hsCRP and leukocytes in middle-aged participants [4]. Moreover, proinflammatory cytokines such as IL-6 and TNFα induce resistin production both ex vivo and in vivo [5] and resistin has also been shown to up-regulate IL-6, TNFα and its own activity via positive feedback [3]. Therefore, resistin could act as both indicator and initiator of the inflammatory process contributing to the development of various chronic diseases, such as atherosclerosis. In fact, recent randomized, double-blinded, clinical trials (CANTOS, COLCOT, LoDoCo2) showed the effects of two anti-inflammatory drugs, canakinumab and colchicine, in reducing cardiovascular events [20–22], thus confirming the role of inflammation in cardiovascular diseases. Previous studies have also documented the association between inflammation and cancer [23] as well as neurodegenerative disorders [24]. In light of these findings, it seems plausible that the association between resistin and increased risk for mortality due to all causes is mediated through its accelerating effect on inflammatory processes. However, our results show that resistin is a risk factor for all-cause mortality independent from inflammatory status (hsCRP). Likewise, Butler et al. [7] reported that the association between resistin concentration and new onset heart failure persisted even after adjusting for CRP, IL-6 and TNFα. Hence, some additional mechanisms not related to proinflammatory properties of resistin should be considered for the related increase in mortality.

Apart from inflammation, resistin has been shown to participate in several other processes promoting cardiovascular and non-cardiovascular morbidity and mortality. In vitro -studies have demonstrated that resistin promotes human aortic smooth muscle cell proliferation [10] and that resistin contributes to endothelial activation promoting atherosclerosis by increasing the expression of endothelin-1, vascular cell adhesion molecule-1 and monocyte chemoattractant protein-1 for example [25]. Moreover, Chen et al. [11] reported that culturing endothelial cells with resistin (40 and 80 ng/ml) for 24 h resulted in significant reduction in endothelial nitric oxide synthase (eNOS) activity, NO bioavailability as well as increase in reactive oxygen species production. The findings of these previous studies support the results of Reilly et al. [12] who showed that in addition to inflammation markers, resistin was associated with coronary artery calcification in middle-aged participants.

Evidence from previous studies also suggest that resistin could be an essential modulator in tumor growth and chemoresistance. Zhang M et al. [26], for instance, showed that injection of resistin significantly accelerated subcutaneous tumor growth and chemoresistance in mice. Similarly, Deshmukh et al. [27] showed that resistin induced breast cancer cell growth and aggressiveness through increased expression and phosphorylation of signal transducer and activator of transcription 3 (STAT3). Finally, resistin could play a role in the pathogenesis of neuroinflammation by increasing the secretion of high mobility group box 1 via activation of p38MAPK and NF-κB signaling pathways [28]. In fact, several previous case-control studies have documented elevated resistin levels in neurodegenerative disorders such as dementia [29], Alzheimer's disease [30] and Multiple Sclerosis [31].

Taken together, resistin is an interesting molecule contributing to various pathophysiological processes which could provide an explanation for resistin-associated cardiovascular and non-cardiovascular comorbidities and mortality. Resistin has also been proposed to be a biomarker of aging [14] and considering its proinflammatory properties, it could be a modulator of the aging-related inflammatory process also known as inflammaging [32]. In light of our results and the previous studies, measuring resistin from elderly patients during clinical

examination could provide useful information regarding the patient's biological age, the on-going inflammatory processes and increased resistin concentration could indicate a higher risk for mortality. Yet, despite 20 years of research, a specific resistin receptor remains to be discovered [1]. Resistin receptor and signaling pathways should be addressed by future studies as possible therapeutic opportunities for delaying or preventing resistin-associated inflammation and comorbidities.

## Conclusions

Regardless of the rich body of research on the biological roles of resistin and its contribution to various chronic disorders, there are not many studies exploring the role of resistin as a risk factor for all-cause mortality among relatively healthy elderly population. Our novel results indicate that, among elderly Finnish participants representing the general population, resistin is a significant risk factor for all-cause mortality independent from traditional cardiovascular risk factors.

## Acknowledgments

The authors thank Ms Saija Kortetjärvi, Ms Heidi Häikiö and Ms Leena Ukkola for the excellent technical assistance.

## Author Contributions

**Formal analysis:** Karri Parkkila.

**Investigation:** Karri Parkkila.

**Methodology:** Antti Kiviniemi.

**Project administration:** Y. Antero Kesäniemi, Olavi Ukkola.

**Supervision:** Mikko Tulppo, Juha Perkiömäki, Olavi Ukkola.

**Writing – original draft:** Karri Parkkila.

**Writing – review & editing:** Antti Kiviniemi, Mikko Tulppo, Juha Perkiömäki, Y. Antero Kesäniemi, Olavi Ukkola.

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
