## [Decision Letter · Decision Letter 0]

13 Jan 2021

PONE-D-20-38390

Resistin is a risk factor for all-cause mortality in elderly Finnish population: a prospective study in the OPERA cohort

PLOS ONE

Dear Dr. Ukkola,

Thank you for submitting your manuscript to PLOS ONE. After careful consideration, we feel that it has merit but does not fully meet PLOS ONE’s publication criteria as it currently stands. Therefore, we invite you to submit a revised version of the manuscript that addresses the points raised during the review process.

Specifically, please improve the data analysis and presentation according to the comments of the two Reviewers. 

We look forward to receiving your revised manuscript.

Kind regards,

Yan Li, MD, PhD

Academic Editor

PLOS ONE

Journal Requirements:

2.) Please include additional information regarding the survey or questionnaire used in the study and ensure that you have provided sufficient details that others could replicate the analyses. For instance, if you developed a questionnaire as part of this study and it is not under a copyright more restrictive than CC-BY, please include a copy, in both the original language and English, as Supporting Information.

3.) We note that you have indicated that data from this study are available upon request. PLOS only allows data to be available upon request if there are legal or ethical restrictions on sharing data publicly. For information on unacceptable data access restrictions, please see http://journals.plos.org/plosone/s/data-availability#loc-unacceptable-data-access-restrictions.

4) Thank you for stating the following financial disclosure:

 [NO. The funders had no role in study design, data collection and analysis, decision to

publish, or preparation of the manuscript.].

Reviewers' comments:

Reviewer's Responses to Questions

**Comments to the Author**

1. Is the manuscript technically sound, and do the data support the conclusions?

Reviewer #1: Partly

Reviewer #2: Partly

2. Has the statistical analysis been performed appropriately and rigorously? 

Reviewer #1: Yes

Reviewer #2: Yes

3. Have the authors made all data underlying the findings in their manuscript fully available?

Reviewer #1: No

Reviewer #2: No

4. Is the manuscript presented in an intelligible fashion and written in standard English?

Reviewer #1: Yes

Reviewer #2: Yes

5. Review Comments to the Author

Reviewer #1: Comments to the paper entitled "Resistin is a risk factor for all-cause mortality in elderly Finnish population: a prospective study in the OPERA cohort"

The authors evaluated a total of 599 Finnish elderly population with a mean follow-up of 5.6 years. They found that resistin is associated with prospective all-cause mortality in this general population of elderly.

The paper should be revised to a major extent, as:

1. No hazard ratio (HR) was provided for the second tertile in the unadjusted model, and all HRs were missing in the adjusted model. They should be listed in a separate TABLE or FOREST PLOT. Moreover, the authors did not adjust for blood pressure as continuous variable(s) in their analysis. Since blood pressure is a significant predictor of cardiovascular and all-cause mortality, it should be added in the model as covariate(s). The necessity was further emphasized in line 280-283, where the authors citied the study by Muse et al. In that study systolic blood pressure was also evaluated by replacing hypertension as sensitivity analysis.

2. Resistin should be log-transformed if the distribution is, as stated in line 86-87, "average resistin concentration was 11.0 ± 5.05 ng/ml and varied from 2.94 to 43.1 ng/ml". Its association with mortality, as revealed in the figure, also supports such transformation that the survival of 1st and 2nd resistin tertile was not of significant difference when inspected visually. Cutoff for resistin, albeit not a must, is also welcomed if provided.

3. Table 1 and 2 were redundant when table 3 presented. They should be removed.

4. The figure should be improved to include p value of log-rank test at the upper-left part and number at risk at the lower part of the figure.

5. Amount of cigarette /alcohol should be described carefully as the distribution of them was largely skewed. The alternative, the proportion of current/history of smoking/heavy drinking, is a better replacement if never-smokers and occasional drinkers constitute a great proportion of the whole.

6. The discussion part should be re-categorized into three major parts, 1. comparison and contrast with previous population study, 2. mechanisms involved in the association, and 3, clinical perspective. In fact, more than half of all your references (Ref #1-8, 10-13, 15, 16, 18, 20, 27, 34, 39-41) came from the first reference. Although citing these papers could enrich your argument on the mechanistic prospective for enabling such association between resistin and all-cause mortality, these arguments and related references should be abbreviated. Only the most important papers relevant to the current findings should be citied.

Some minor typing errors/suggestions:

line 22 "monoculear" should be "mononuclear"

line 57 a comma "," should be added after ref [11]

line 134 a comma "," should be added after "total".

line 136-137 the concentration starting with (1.63...) should follow HDL but not cholesterol.

line 137 "represented" should be "presented"

line 238 "inceased" should be "increased"

line 289 "spesific" should be "specific"

Reviewer #2: In this manuscript, the author investigated that whether resistin predicts mortality among elderly Finnish people. The study population consisted of 599 elderly (71.7 ± 5.4 years) patients and the follow-up was approximately six years. After the follow-up, resistin was a significant risk factor for all-cause mortality (HR 3.02, 95 % CI: 1.64-5.56, p<0.001) even after adjusting for various covariates such as age, sex, hypertension, diabetes, hsCRP, physical activity, smoking and alcohol consumption. The result indicated that resistin is a significant risk factor for all-cause mortality among elderly Finnish subjects, independent from traditional cardiovascular risk factors. However, there are several issues need to be considered.

1、 How about the distribution of resistin in the population? Do you need to use the log transformed value.

2、 This study demonstrated that resistin was a significant risk factor for all-cause mortality when analyzed as a continuous variable as well as when the highest and lowest tertile were compared. It would be better if you do Receiver operator characteristic (ROC) curve analysis for serum resistin to determine the possible cutoff value for resistin.

3、 This study demonstrated that resistin was a significant risk factor for all-cause mortality. How about the CV mortality? Do you have the related data?

4、 I wonder whether or not the highly increased total mortality risk was modulated by synergistic interaction with various demographic and clinical features. To this purpose, mortality risk should be compared across subgroups in a multivariable model considering general confounders, such as sex, age at recruitment, smoking habits and so on.

5、 In Figure 1, the accurate resistin concentration in different tertiles and the P value should be illusterated.

6、 Univariate Cox-regression analysis and multivariate Cox regression analysis should be illustrated in Table in different models. Glucose-lowering, anti-hypertensive, antidyslipidemic treatments should be adjusted in multivariate Cox regression analysis.

7、 What’s the device you used in this study to measure blood pressure? And how many times you measured to define hypertension? What’s the definition of diabetes? You should introduce EQ-5D and MMSE briefly in the Methods.

8、 In line 126-129, you mentioned that :the discrimination abilities of resistin were assessed by the C-index by adding the following covariates into established model: age, hypertension, diabetes, hsCRP, physical activity and resistin. However, I did see any result about the C-index in this manuscript. Maybe you’d better add ROC curve in your manuscript.

6. PLOS authors have the option to publish the peer review history of their article (what does this mean?). If published, this will include your full peer review and any attached files.

Reviewer #1: No

Reviewer #2: No

---

## [Author Response · Author response to Decision Letter 0]

28 Jan 2021

All the respective comments of both reviewers and the editor have been addressed in the attached "Response to Reviewers" file.

---

## [Decision Letter · Decision Letter 1]

18 Feb 2021

Resistin is a risk factor for all-cause mortality in elderly Finnish population: A prospective study in the OPERA cohort

PONE-D-20-38390R1

Dear Dr. Ukkola,

We’re pleased to inform you that your manuscript has been judged scientifically suitable for publication and will be formally accepted for publication once it meets all outstanding technical requirements.

Kind regards,

Yan Li, MD, PhD

Academic Editor

PLOS ONE

Additional Editor Comments (optional):

Reviewers' comments:

Reviewer's Responses to Questions

**Comments to the Author**

1. If the authors have adequately addressed your comments raised in a previous round of review and you feel that this manuscript is now acceptable for publication, you may indicate that here to bypass the “Comments to the Author” section, enter your conflict of interest statement in the “Confidential to Editor” section, and submit your "Accept" recommendation.

Reviewer #1: All comments have been addressed

Reviewer #2: All comments have been addressed

2. Is the manuscript technically sound, and do the data support the conclusions?

Reviewer #1: (No Response)

Reviewer #2: Yes

3. Has the statistical analysis been performed appropriately and rigorously? 

Reviewer #1: (No Response)

Reviewer #2: Yes

4. Have the authors made all data underlying the findings in their manuscript fully available?

Reviewer #1: (No Response)

Reviewer #2: Yes

5. Is the manuscript presented in an intelligible fashion and written in standard English?

Reviewer #1: (No Response)

Reviewer #2: Yes

6. Review Comments to the Author

Reviewer #1: (No Response)

Reviewer #2: (No Response)

7. PLOS authors have the option to publish the peer review history of their article (what does this mean?). If published, this will include your full peer review and any attached files.

Reviewer #1: No

Reviewer #2: No

---

## [Editor Report · Acceptance letter]

22 Feb 2021

PONE-D-20-38390R1 

Resistin is a risk factor for all-cause mortality in elderly Finnish population: A prospective study in the OPERA cohort 

Dear Dr. Ukkola:

I'm pleased to inform you that your manuscript has been deemed suitable for publication in PLOS ONE. Congratulations! Your manuscript is now with our production department. 

Kind regards, 

on behalf of

Professor Yan Li 

Academic Editor

PLOS ONE